# Actual and Potential Role of Primary Care Physicians in Cancer Prevention

**DOI:** 10.3390/cancers15020427

**Published:** 2023-01-09

**Authors:** Marta Mańczuk, Irena Przepiórka, Magdalena Cedzyńska, Krzysztof Przewoźniak, Elwira Gliwska, Agata Ciuba, Joanna Didkowska, Paweł Koczkodaj

**Affiliations:** Cancer Epidemiology and Primary Prevention Department, Maria Sklodowska-Curie National Research Institute of Oncology, 02-781 Warsaw, Poland

**Keywords:** cancer prevention, primary care physicians, Poland, European Code Against Cancer

## Abstract

**Simple Summary:**

Cancer remains one of the main public health concerns worldwide. Up to 50% of all cancers can be prevented by following lifestyle recommendations and health professional advice. The role of Primary Care Physicians (PCPs) is invaluable when it comes to primary and secondary cancer prevention efficiency, not only in the context of tobacco prevention and smoking cessation, but also in the case of other cancer risk factors described in the European Code Against Cancer (ECAC), such as obesity, poor diet, low physical activity, alcohol consumption or non-participation in screenings. Understanding the actual and potential role of PCPs, as it is perceived by themselves, creates an opportunity to face challenges and improve percentages of cancer prevention advice received in the primary care setting. The study findings bring valuable insight into the actual and potential role of PCPs in cancer prevention in delivering knowledge and convincing patients that their health depends on themselves and their lifestyle choices.

**Abstract:**

Although the role of primary care in cancer prevention has been proven, its assumptions are still insufficiently implemented and the actual rates of cancer prevention advice delivery remain low. Our study aimed to identify the actual and potential role of primary care physicians (PCPs) in the cancer prevention area. Design of the study is a cross-sectional one, based on a survey of 450 PCPs who took part in a nationwide educational project in Poland. Only 30% of PCPs provide cancer prevention advice routinely in their practice, whereas 70% do that only sometimes. PCPs’ actual role in cancer prevention is highly unexploited. They inquire routinely about the patient’s smoking history (71.1%), breast cancer screening program (43.7%), cervical cancer screening (41.1%), patient’s alcohol consumption (34%), patient’s physical activity levels (32.3%), body mass index (29.6%), the patient’s eating habits (28%) and patient’s potential for sun/UV-Ray exposure (5.7%). The potential role of PCPs in cancer prevention is still underestimated and underutilized. Action should be taken to raise awareness and understanding that PCPs can provide cancer prevention advice. Since lack of time is the main obstacle to providing cancer prevention advice routinely, systemic means must be undertaken to enable PCPs to utilize their unquestionable role in cancer prevention.

## 1. Introduction

Cancer remains one of the main public health concerns worldwide. At the same time, up to 50% of all cancers can be prevented by following lifestyle recommendations and health professional advice [1]. According to *Globocan* estimates, in 2020, more than 18 million people developed cancer (Age Standardized Rate—ASR: 190.0/100,000), and almost 10 million men and women died due to this disease (ASR: 100.1) [2]. It has been estimated that in 2040, probable global cancer burden will reach the level of 28 million new cases (close to 50% increase in comparison to 2020) [3]. 

In only 27 European Union (EU) countries in 2020, there were almost 2.7 million (ASR: 283.6/100,000) new cancer cases and more than 1.2 million cancer deaths in both sexes (ASR: 106.7). Among EU-27 females, the most frequent cancer sites were: breast: (28.7%) cases (ASR: 82.8); colon and rectum: 150,366 (12.2%) (ASR: 24.7); and lung: 113,074 (9.1%) (ASR: 21.3). In males—prostate: 335,514 (23.2%) cases (ASR: 69.5); lung: 205,253 (14.2%) (ASR: 43.0); and colorectal: (13.2%) (ASR: 38.8). The most frequent causes of cancer deaths were not an exact reflection of the aforementioned incidence data, both for women and men. The highest proportion of European females died due to lung cancer: 86,731 (15.6%) cases (ASR: 15.2); breast cancer: 91,826 (16.5%) (ASR: 14.0); and colon and rectum cancer: 68,920 (12.4%) (ASR: 8.8). In the population of European males, the three most frequent causes of cancer deaths were as follows: lung cancer: 170,562 (24.2%) cases (ASR: 34.2); colorectal cancer: 87,185 (12.3%) (ASR: 15.5); and prostate cancer: 69,945 (9.9%) (ASR: 10.2) [4]. 

In Poland, many highly preventable cancers (taking into account primary and secondary cancer prevention actions) are among the most frequent ones. The most tangible example in this matter is lung cancer, wherein about 85% of cases are caused by a single, highly avoidable risk factor—tobacco smoke [5]. Considering the latest data from the Polish National Cancer Registry, in 2019, lung cancer was the second most frequent one among females and males (right after breast cancer and prostate cancer) with the respective number of cases at the level of 8469 (ASR: 19.2) and 13 802 (ASR: 39.3). On the other hand, lung cancer was the first cause of cancer deaths in Poland, in both sexes, contributing in 2019 to 8205 (ASR: 17.6) deaths in Polish females and as many as 14,902 (ASR: 41.7) in males [6]. This information is of importance because the latest data on smoking prevalence in Poland suggest a significant increase in the percentage of current smokers (28.8% of current smokers aged 15 and older in 2022 vs. 26% in 2020 and 22.3% in 2019) [7]. Moreover, the health consequences of tobacco smoke exposure are much more far-reaching than the increased risk of lung cancer solely [8].

The role of primary care physicians (PCPs) is invaluable when it comes to primary and secondary cancer prevention efficiency, not only in the context of tobacco prevention and smoking cessation, but also in the case of other cancer risk factors described in the European Code Against Cancer (ECAC), including, but not limited to, obesity, poor diet, low physical activity, alcohol consumption or non-participation in screenings [9]. Moreover, there is evidence that the role of health professionals, including PCPs, plays a key role in increasing screening participation rates, which, in Poland, are exceptionally low [10]. 

Our study aimed to identify the actual and potential role of PCPs in the cancer prevention area. Outcomes will bring new light to cancer prevention activities in primary health settings and may result in actions to increase cancer prevention’s role in the public health sector, which, in a further perspective, may also contribute to lowering the cancer burden in Poland. 

## 2. Materials and Methods 

### 2.1. Participants

Participants of this study were recruited from the general population of PCPs and other medical doctors (employed in primary care settings) who accessed the project called “Cancer Vigilance in Primary Healthcare: Nationwide Series of Courses on Primary Prevention, Screening and Dealing With Anticancer Treatment Complications and Cancer Pain” funded by the EU from the European Social Fund (grant no POWR.05.04.00-00-00-0068/16-00/97/2017/2/45). The grant was awarded to the Maria Sklodowska-Curie National Research Institute of Oncology in Warsaw, Poland, and implemented during 2017–2019. Inclusion criteria were set as a medical doctor who is employed in a primary care setting and provides primary care service (mostly with a specialty of primary care or internal medicine. For details refer to Table 1).

### 2.2. Data Collection

During the course of the abovementioned project, the participants were given a set of questionnaires that included questions about their actual and potential role in cancer prevention. The main questionnaire was adopted from Macilfatric and colleagues [11]. There were, in total, 38 questions regarding their actual role in cancer prevention and 7 in-depth questions regarding their opinion about their potential role in cancer prevention. The questionnaires were anonymous. There was a total of 450 questionnaires collected; for some parts of the questionnaire, especially the second in-depth part, there were 380 returns.

### 2.3. Data Analysis

After the collection of the questionnaires, the data were cleaned and checked for inconsistencies. Descriptive statistics were calculated using Microsoft Excel and SAS software. Z-score was used to test the significance of differences between groups.

### 2.4. Limitations of the Study

The declarative character of collected data is a potential limitation of our study. However, as all participants filled out the survey anonymously and its topic was not perceived as a controversial one, we can assume high accuracy of the given answers. 

Moreover, in our study, we obtained a feminized sample—71% females vs. 29% males, which could potentially affect the overall character of answers. On the other hand, the overall number of medical doctors (professionally active) in Poland shows that overproportion of women in this occupational group is visible also at the national level—at the end of November 2022, there were 62,159 male medical doctors vs. 88,733 female doctors [12]. We assumed that our sample was affected by this regularity to some extent.

Not all of the questionnaires were filled completely (up to 6 missing records in various categories), therefore not all categories sum up to the primary sample size of *N =* 450. 

## 3. Results

### 3.1. Demographics

In our study, the highest proportion of participants was represented by females—71% (300) (71%) vs. 29% (males). PCPs were the most represented professional group—64% (274). Respondents in our sample were in majority young 37% (163) or middle-aged, with the highest percentage of those at the age of 35–59 years old—55% (245) (Table 1). 

### 3.2. Actual Role in Cancer Prevention

One of the most crucial areas, investigated in our study, was identification of the actual role of PCPs in the cancer prevention area. About 1/3 of them—28.7% (108)—provide cancer prevention advice routinely in their practice. Further, only 7.8% (29) of PCPs claimed that routinely provide leaflets/information sheets relating to cancer prevention to patients. Considering smoking cessation services, as much as 71.1% of doctors claimed to enquire about a patient’s smoking habits/history routinely (the most frequent cancer prevention service provided routinely by PCPs). On the other hand, not many of them inform regularly about treatment options—25.2%, in the case of Nicotine Replacement Therapy (NRT), and 22% for cytisine. Moreover, the prescription of smoking cessation drugs during appointments seems to be not very often practiced—only 8.1% (30) of doctors do this routinely. Discussing services related to obesity, for almost 1/3 of interviewed doctors—29.6% (110)—measuring patients’ weight/height/body mass index was a routine action. In the context of obesity prevention, the most frequent action declared as a regular practice was weight management advice—48% (177) of respondents. Apart from that, only 10.5% (39) of doctors provide patients routinely with materials about the connection between obesity and increased cancer risk. In the case of physical activity, 32.3% (119) of respondents always enquire about the patient’s physical activity levels, which, in comparison with smoking behaviors, was a much lower percentage (32.3 vs. 71.1%). Similar to obesity, also in the physical activity case, the percentage of doctors who routinely give patients leaflets or information sheets about the relation between cancer risk and levels of physical activity was low—5.1% (19). Moreover, a very low proportion of PCPs who routinely provide written information on cancer prevention was characteristic for categories such as diet—8.8% (33); alcohol consumption—5.4% (20) and UV exposure—as low as 3.5% (13). A slightly higher percentage was characteristic for the last category—screenings—12.5% (46). However, considering this category, we investigated that 43.7% (162), 41.1% (153) and 31.4% of doctors enquire routinely about patients’ participation in breast, cervical and colorectal cancer screenings, respectively. 

In general, irrespective of the analyzed category, the vast majority of doctors chose the answer “sometimes”, which may suggest that during clinical visits, cancer prevention is not necessarily one of their priorities. Moreover, in many cases answer “not at all” was also quite frequent (e.g., provision of printed information materials), which indicates that the actual role of doctors in cancer prevention could be potentially much stronger (Table 2).

### 3.3. Barriers to Cancer Prevention Role

The studied group of PCPs was also asked for reasons if they indicated the answer “not at all”. The main reasons they named were a lack of time (from 12 to 100% in particular categories), lack of financial resources (from 6 to 30%) and other unspecified reasons (from 12 to 50%). 

### 3.4. The potential Role of Primary Care Physicians in Cancer Prevention

In the next part of the study, we asked participants about their opinion on the potential role of the PCPs in the cancer prevention area (Table 3). As much as 67.7% (252) of respondents strongly agreed with the opinion that empowering patients to make their own health decisions is the most important potential doctor’s role (the most frequent answer, regardless of agreement or disagreement level). Additionally, 31.5% (117) of them agreed that empowerment plays an important role as well; however, this group chose the answer “agree” instead of “strongly agree”. The next group, characterized by the highest proportion of “strongly agree” answers, indicated the category “Identifying patients at risk”—62% (230). Stronger agreement was also characteristic for categories such as: “Offering advice to inform individuals about better lifestyle choices” and “Ensuring equality of access to cancer prevention interventions”—respectively 59.1% (221) and 56.% (208) of doctors strongly agreed with these potential roles. The lowest agreement occurred in categories: “Working with local communities to empower them to make decisions about lifestyle choices”—32.5% (121) respondents strongly agreed vs. 44.6% (166) who just agreed (also the highest percentage of “no opinion” answers—21% (78) in this category) as well as “Ensuring a coordinated cancer prevention approach within the practice”—50% (186) vs. 44.8% (167), respectively, strongly agreed and agreed on the crucial meaning of this potential role (Table 3). 

### 3.5. Perceived Responsibility, Knowledge and Acceptability of a Primary Care Physician in Relation to Cancer Prevention Role

In the last part of the study, we asked respondents about perceived responsibility, knowledge and acceptability of a Primary Care Physician in relation to the cancer prevention role. In relation to the first category—responsibility—the highest percentage of doctors claimed that PCPs should try and provide cancer prevention services—46.9% (173) and 51% (188) of them, respectively, strongly agreed and just agreed with this statement. As much as 20.8% (76) of doctors strongly agreed that PCPs should screen high-risk cancer groups, which gives an interesting inside into the actual perception of their role. Moreover, the highest disagreement percentage for discussed category (“responsibility”) occurred in the case of a statement linked to the time doctors devote to cancer treatment vs. cancer prevention services. A total of 17.3% (64) of them disagreed that PCPs spend too much time on treatment in comparison to prevention. 

Considering the next category—“knowledge”—only 11.1% (40) of asked doctors strongly agreed that they possess sufficient knowledge to educate patients about cancer prevention. On the other hand, as much as 64.5% (234) agreed with the above statement; however, this choice could suggest some gaps in their knowledge. Additionally, disagreement and strong disagreement occurred in the case of 6.9% (25) and 0.3% (1) of respondents. Further, as many as 25.1% (91) and 65.6% (238) of doctors strongly agreed and just agreed that they require up-to-date information on cancer prevention strategies. Similarly, a high level of agreement was visible for the statement on the need for a better understanding of how to change opinions regarding cancer prevention—19.3% (70) and 54.8 (199) of doctors strongly agreed and agreed. 

The last category—“Perceived Acceptability”— is characterized by relatively low levels of agreement with presented cancer prevention statements (all of them refer to probable patients’ behaviors; therefore, discussed category could possibly provide very valuable indicators of patients’ attitudes as well). The lowest level of agreement occurred in the case of a statement referring to lack of follow-up from the patient’s side after cancer prevention consultation—only 4.1% (15) and 18.8% (69) strongly agreed and agreed with this statement. Furthermore, the very low level of agreement was characteristic of the statement on the positive correlation between patients’ anxiety and cancer prevention intervention. In total, 4.9% (18) and 25.3% (93) of doctors strongly agreed and just agreed, respectively, with this statement (Table 4).

## 4. Discussion

In our study, the majority of PCPs (98%) clearly see their role in cancer prevention, by empowering patients to make their own decisions about health issues and offering them advice to facilitate better lifestyle choices. Slightly lower rates were published by McIlfatric and colleagues [11], where 93% of PCPs acknowledge that they should try and provide cancer prevention services. That is in the contrast to the findings of our study that only 30% of PCPs are actually providing cancer prevention advice routinely. Among British General Practitioners, GPs, this percentage was twice as high (66.4%) [11]. In another study, performed on nurses, the percentage of providing general cancer prevention services routinely was close to that observed in GPs (59.6%) [13]. Ngwakognwi and colleagues in another study were investigating medical documentation run by GPs, looking for any cancer preventive interventions [14]. Their results were similar to the findings of this study, but only when documented cancer prevention intervention happened within 1 year (up to 40%), whereas when the time range was spread for 3 years to intervention rate grew up to 67%. Another study, however, reports that over 90% of medical doctors routinely provide preventive screenings and interventions to their patients [15]. It would be interesting to see what the reasons behind such differences in the study results are regarding routinely providing cancer prevention services. 

Undoubtedly PCPs are at the front line of healthcare services and have important roles in primary prevention and screening for cancer [16]. However, our findings and other literature on the subject indicate that the actual role of PCPs is differentiated within the spectrum of cancer prevention services for particular risk factors and particular cancers. When going through Canadian GPs’ medical charts, Ngwakognwi and colleagues discovered that the highest rate of documented need for intervention was for cholesterol measurement (within 1 year), 40.3%, then for mammography, 28.2%, and cytology, 14.8%. Surprisingly, smoking cessation service was offered only in 3% of eligible charts [13]. They discovered also a dangerous practice of offering unnecessary cancer prevention service, as for the Prostate-Specific Antigen (PSA) test, 7.5%, and digital rectal examination, 18.9%. 

Studies that presented data from surveys reported rates of particular cancer prevention services that PCPs provide routinely to their patients. Our findings show that the provision was highest for smoking cessation (71.1%), whereas it was 96.8% among British GPs [11] and 96% among US PCPs [14]. Over 40% of PCPs in our study provided advice on mammography (43.7% vs 98% among the US PCPs [14]) and cytology screening (41.1% vs 95.3% among British GPs [11] and 94% in the US [14]). Further, findings from our study are generally lower than reported by other studies: alcohol consumption (34% vs. 71.7% [11]), physical activity levels (32.3% vs. 55.8% [11]), weight management (29.6% vs. 77.8% [11]), diet (28% vs 55.4% [11] and 81.6% [14]) and UV exposure (5.7% vs. 17.9% [11]). The reasons for these discrepancies should be investigated in further research. One of the explanations might be the study design. The limitations of self-report surveys need to be acknowledged. However, given that the study participants were recruited from an already selected group of PCPs, who participated in an educational project, it is possible that the PCPs were accurately reporting their actual practice. However, in studies with surveys performed without the context, reported rates might reflect the best practice of PCPs rather than what they actually do.

The study by Amelung et al. [17] showed that the appropriate doctor–patient relationship and early symptoms awareness in primary care could affect the timeliness of cancer detection and subsequent treatment effectiveness. Moreover, according to Harris et al. [18] well-educated clinicians and adequate funding for primary care cancer diagnostic pathways may influence more timely cancer diagnosis. Cancer prevention support also seems to be important for patients and many studies have confirmed that patients are willing to discuss their cancer risk with their primary care physician. Tackling difficulties and barriers among PCPs might be an effective way of improving cancer prevention intervention effectiveness; however, many studies similar to this research indicate inadequate cancer prevention practice implementation.

The main obstacles indicated by PCPs in our study include lack of time and lack of financial resources for providing cancer prevention services, which is consistent with other studies [11,14]. Many PCPs in our study declared that they have other reasons for not providing a particular service. These other reasons require further investigation.

The findings regarding the potential role of PCPs in cancer prevention are quite consistent with the results obtained by Mcilfatrick et al. [11] among British GPs. Both Polish and British first-contact medical doctors strongly agreed (67.7% vs. 64.1%, *p* for difference > 0.05) or just agreed (31.5% vs. 34.8%, *p* > 0.05) that their role should be about empowering individuals to make their own decisions about health issues. With the statement that their cancer prevention role should be about offering advice to inform individuals about better lifestyle choices, 59.1% vs. 66.8% of Polish and British PCPs strongly agreed, respectively, (*p* for difference > 0.05), or just agreed 38.2% vs. 32.1% (*p* for difference > 0.05). Interestingly, findings of this study revealed that a significantly higher percentage of PCSs see that their cancer prevention role should be also about working with local communities to empower them to make decisions about lifestyle choices than in the Mcilfatric et al. study [11]: 32.5% vs. 22.3% (*p* for difference < 0.05) strongly agreed with the statement; there was no significant difference between those who just agreed (44.6% vs. 39.9%, *p* > 0.5). Many studies show that community engagement strategies are helpful in enrolling diverse populations into cancer prevention services [19,20], especially those underserved and often from the higher cancer risk groups [21,22]. In this context, it would be interesting to explore this issue further.

## 5. Conclusions

Only 30% of PCPs provide cancer prevention advice routinely in their practice, whereas 70% do that only sometimes.

PCPs’ actual role in cancer prevention is highly unexploited. They inquire routinely about patients’ smoking history (71.1%), participation in the breast cancer screening program (43.7%), participation in the cervical cancer screening (41.1%), patients’ alcohol consumption patterns (34%), patient’s physical activity levels (32.3%), measure weight/height/body mass index (29.6%) and enquire about patients’ eating habits (28%). Only 5.7% of PCPs routinely inquire about patients’ potential for sun/UV-Ray exposure.

The results of our study suggest that cancer prevention is not necessarily PCPs’ priority during appointments. There is a need for systemic changes to strengthen cancer prevention place in the medical studies curriculum.

The main obstacles indicated by PCPs in our study include lack of time and lack of financial resources.

The potential role of PCPs in cancer prevention is underutilized also by patients’ approaches and attitudes. Action should be taken to raise awareness and understanding that PCPs can provide cancer prevention advice. On the other hand, since lack of time is the main obstacle to providing cancer prevention advice routinely, systemic means must be taken to help PCPs to utilize their unquestionable role in cancer prevention.

## Figures and Tables

**Table 1 cancers-15-00427-t001:** Main characteristics of the sample.

Category	%	*N **
Female	71%	300
Male	29%	122
Primary Care Physician	64%	274
Internal medicine specialty	24%	103
Other specialties	12%	51
25–34 years old	37%	163
35–59 years old	55%	245
60 years old and more	8%	36

* *N* = 450, categories do not sum up due to missing data.

**Table 2 cancers-15-00427-t002:** Actual role in cancer prevention by chosen risk factors.

Actual Role in Cancer Prevention							Routinely		Sometimes		Not at All	
							*N*	%	*N*	%	*N*	%
Do you provide cancer prevention advice to your patients during visits?							108	28.7%	264	70.0%	5	1.3%
Do you provide general information materials about cancer prevention?							29	7.8%	293	78.6%	51	13.7%
Services related to smoking												
Do you enquire about a patient’s smoking history and habits?							268	71.1%	106	28.1%	3	0.8%
Do you provide brief advice?							201	53.3%	172	45.6%	4	1.1%
Do you provide specialist services?							49	13.0%	274	72.7%	54	14.3%
Do you advise using Nicotine Replacement Therapy?							94	25.2%	265	71.1%	14	3.8%
Do you advise using OTC drugs helping to quit smoking (cytisine)?							82	22.0%	272	72.9%	19	5.1%
Do you prescribe smoking cessation drugs (bupropion, varenicline)?							30	8.1%	277	74.7%	64	17.3%
Do you provide information materials about smoking/passive smoking and cancer risk?							48	12.9%	230	61.7%	95	25.5%
Do you refer patients to other services, such as a smoking cessation clinic?							26	7.1%	231	62.9%	110	30.0%
Services related to obesity												
Do you measure patients’ weight and height to calculate body mass index?							110	29.6%	245	65.9%	17	4.6%
Do you provide information materials about excess body mass and cancer risk?							39	10.5%	230	62.0%	102	27.5%
Do you display Height/Weight/Body Mass Index Charts in public areas within the Practice?							68	10.5%	58	15.8%	242	65.8%
Do you provide weight management advice?							177	48.0%	187	50.7%	5	1.4%
Do you refer patients to other services?							86	23.2%	267	72.2%	17	4.6%
Services related to physical activity												
Do you enquire about a patient’s physical activity levels?							119	32.3%	239	64.8%	11	3.0%
Do you provide information materials containing the requirement for daily physical activity?							39	10.4%	243	65.0%	92	24.6%
Do you provide information materials about physical activity and cancer risk?							19	5.1%	233	62.8%	119	32.1%
Do you refer patients to other services?							81	21.8%	279	75.2%	11	3.0%
Services related to diet												
Do you enquire about a patient’s eating habits?							105	28.0%	258	68.8%	12	3.2%
Do you provide information materials about the relationship between diet and cancer?							33	8.8%	266	71.1%	75	20.1%
Do you provide information materials about eating at least 5 servings of fruit and vegetables daily?							54	14.5%	272	72.9%	47	12.6%
Do you refer patients to other services?							38	10.3%	305	82.7%	26	7.1%
Services related to alcohol												
Do you enquire about a patient’s alcohol consumption?							127	34.0%	241	64.4%	6	1.6%
Do you provide information materials about the consumption of alcohol?							20	5.4%	217	58.3%	135	36.3%
Do you provide information materials about alcohol consumption and cancer risk?							21	5.7%	223	60.0%	128	34.4%
Do you refer patients to other services?							68	18.2%	288	77.0%	18	4.8%
Services related to sun/UV exposure												
Do you enquire about a patient’s potential for exposure to UV radiation?							21	5.7%	267	72.4%	81	22,0%
Do you provide information materials about UV radiation exposure and cancer risk?							13	3.5%	201	54.5%	155	42.0%
Do you provide information materials about protection from UV rays, necessary for fair skin?							12	3.3%	189	51.8%	164	44.9%
Do you refer patients to other services?							65	17.7%	287	78.0%	16	4.4%
Services related to cancer screening programs												
Do you enquire about patients’ participation in the cervical cancer screening program?							153	41.1%	194	52.2%	25	6.7%
Do you provide information materials about cancer screening programs and cancer risk?							46	12.5%	223	60.6%	99	26.9%
Do you refer patients to other services?							160	43.1%	203	54.7%	8	2.2%
Do you enquire about patients’ participation in the breast cancer screening program?							162	43.7%	194	52.3%	15	4.0%
Do you enquire about patients’ participation in the colorectal cancer screening program?							116	31.4%	231	62.4%	23	6.2%

**Table 3 cancers-15-00427-t003:** Potential role of Primary Care Physicians in the prevention of cancer.

Potential Role of PCPs *	Strongly Agree	Agree	No Opinion	Disagree	Strongly Disagree
Empowering individuals to make their own decisions about health issues	67.70%	31.50%	0.80%	0%	0%
	(*n* = 252)	(*n* = 117)	(*n* = 3)	(*n* = 0)	(*n* = 0)
Offering advice to inform individuals about better lifestyle choices	59.10%	38.20%	2.10%	0.50%	0%
	(*n* = 221)	(*n* = 143)	(*n* = 8)	(*n* = 2)	(*n* = 0)
Working with local communities to empower them to make decisions about lifestyle choices	32.50%	44.60%	21%	1.60%	0.30%
	(*n* = 121)	(*n* = 166)	(*n* = 78)	(*n* = 6)	(*n* = 1)
Ensuring a coordinated cancer prevention approach within the practice	50%	44.80%	4.30%	1.10%	0%
	(*n* = 186)	(*n* = 167)	(*n* = 16)	(*n* = 4)	(*n* = 0)
Identifying patients at risk	62%	35.60%	1.90%	0.50%	0%
	(*n* = 230)	(*n* = 132)	(*n* = 7)	(*n* = 2)	(*n* = 0)
Ensuring equality of access to cancer prevention interventions	56.10%	38%	4.30%	1.10%	0.50%
	(*n* = 208)	(*n* = 141)	(*n* = 16)	(*n* = 4)	(*n* = 2)

* Question: “As a Primary Care Physician, I feel that my cancer prevention role should be about:”.

**Table 4 cancers-15-00427-t004:** Perceived responsibility, knowledge and acceptability of a Primary Care Physician in cancer prevention role.

**Responsibility**	**Strongly Agree**	**Agree**	**No Opinion**	**Disagree**	**Strongly Disagree**
Primary Care Physicians should try and provide cancer prevention	46.90%	51%	1.60%	0.50%	0%
	(*n* = 173)	(*n* = 188)	(*n* = 6)	(*n* = 2)	(*n* = 0)
Primary Care Physicians spend too much time on the treatment of cancer rather than providing cancer prevention	15.10%	44%	21.90%	17.30%	1.60%
	(*n* = 56)	(*n* = 163)	(*n* = 81)	(*n* = 64)	(*n* = 6)
Primary Care Physicians have a responsibility to screen high-risk cancer groups	20.80%	59.40%	12.10%	7.70%	0%
	(*n* = 76)	(*n* = 217)	(*n* = 44)	(*n* = 28)	(*n* = 0)
**Knowledge**	**Strongly Agree**	**Agree**	**No Opinion**	**Disagree**	**Strongly Disagree**
I have sufficient knowledge to educate patients about cancer prevention	11.10%	64.50%	17.40%	6.90%	0.30%
	(*n* = 40)	(*n* = 234)	(*n* = 63)	(*n* = 25)	(*n* = 1)
I require up-to-date information on cancer prevention strategies	25.10%	65.60%	5.50%	3.10%	0.80%
	(*n* = 91)	(*n* = 238)	(*n* = 20)	(*n* = 11)	(*n* = 3)
I require a better understanding of how to change opinions regarding cancer prevention	19.30%	54.80%	13.50%	11.60%	0.80%
	(*n* = 70)	(*n* = 199)	(*n* = 49)	(*n* = 42)	(*n* = 3)
**Perceived Acceptability**	**Strongly Agree**	**Agree**	**No Opinion**	**Disagree**	**Strongly Disagree**
Patients are very set in their ways and do not want to change	13.30%	51.10%	11.70%	23.90%	0%
	(*n* = 49)	(*n* = 188)	(*n* = 43)	(*n* = 88)	(*n* = 0)
Patients do not like the Primary Care Physician to meddle in their private life	8%	41.90%	18.10%	31.80%	0.30%
	(*n* = 29)	(*n* = 153)	(*n* = 66)	(*n* = 116)	(*n* = 1)
Patients do not approach their Primary Care Physician for advice on cancer prevention	7.90%	40.20%	12.10%	38.30%	1.60%
	(*n* = 29)	(*n* = 147)	(*n* = 44)	(*n* = 140)	(*n* = 6)
Primary Care Physicians may increase anxiety in the patient population by undertaking cancer prevention activities	4.90%	25.30%	15.50%	46.60%	7.60%
	(*n* = 18)	(*n* = 93)	(*n* = 57)	(*n* = 171)	(*n* = 28)
After consultation with a patient on cancer risk, I do not think they will follow my recommendation	4.10%	18.80%	22.60%	51.80%	2.70%
	(*n* = 15)	(*n* = 69)	(*n* = 83)	(*n* = 190)	(*n* = 10)

## Data Availability

Data are available upon reasonable request from the corresponding author.

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
