# Peer review of "Actual and Potential Role of Primary Care Physicians in Cancer Prevention"

_cancers, 2023, doi:10.3390/cancers15020427_

Round 1

Reviewer 1 Report

The role of primary care physicians is extremely important in oncology prevention, not only because of their closest contact with patients, but also because of their ability to provide information about screening and education about oncology prevention. Not only handing over the oncology fast track card so-called DILO card, but also educating and supporting the patient in prevention should be one of the priorities in the work of primary care physicians.

In section data analisys you should describe the data analysis methodology in more detail. this section is too general, it does not bring information on the statistics used.

The tables are prepared in an accurate, clear manner. Well described.

Barriers and facilitators to cancer prevention role - you should describe this section better, with more details. This is very interesting section and if we knwo why physicians don't promote cancer prevention among their patients we will be able to act systemically to change this.

Discussion and conclusions formulated correctly.

Bibliography should be prepared according to the guidelines.

Author Response

Dear Reviewer,

Please find the attached file with our answers.

Thank you,

Authors

Reviewer 2 Report

This is a cross-sectional study to identify the roles primary care physicians play in cancer prevention through patient education and other interventions.

1. Line 27,28,29 : Rephrase this sentence- This is a cross-sectional, paper-based survey of 450 primary care physicians who took part in a nationwide educational project in Poland. 

2. Line 35, 36: Rephrase this sentence.

3. Line 52-64 : In this paragraph, if you can also include percentages of cases, it will be more helpful to quickly understand the impact of these cancers. 

4. Units used throughout the paper: In some places, you have used decimal separator as decimal point (e.g. 283.6) and at other places you have used decimal comma (e.g. 2,7 million). Kindly use a single form throughout the paper.

5. Line 76 : In this sentence, the word "meaningful" can be replaced with "significant".

6. Line 77 : rephrase "15 years and older"

7. Line 84 : you can write "including but not limited to.."

8. Line 90,91 : Rephrase the line. 

9. Line 91 : Obtained outcomes can be simply written as outcomes of our study...

10. Line 98, 99 : Participants of the study - Primary care physicians and other medical doctors - the study specifically determined the impact of primary care physicians. Therefore, it is important to describe in detail the exact "inclusion and exclusion criteria" of study participants.

11: Kindly describe in short who qualifies as a primary care physician in Poland. It will be helpful to mention the qualifications required to work as a primary care physician in Poland.

12:  Throughout the paper, you have used Roman numericals as citation numbers. Even though it is correctly written and an acceptable way of citing articles, if you can use numbers (1,2,3..) it will be easier for the readers to locate them on the paper for their reference.

13: The paper numbering is incorrect. After Page 4 of 13, Table 2 comes, and after that again, Page 1 of 13 begins. Please correct the numbering. 

14: Line 187 : "lack of financial resources" instead of " a lack of financial resources".

15: In table 4, the question appears as " Primary care physicians spend too much time on treatment of cancer rather than providing cancer prevention". Can you please describe what treatment the primary care physicians deliver to cancer patients? 

16: Line 261, 263: Please write the full form of GPs and SPs respectively as it appears for the first time in the paper.

17. Line 277: PCPs - When this abbreviation appears for the first time in the paper, please write primary care physicians (PCPs).

18. Also throughout the study, you used "primary care physician" words in a significant manner. Using PCPs abreviation can make it easier for the readers and also help to not sound monotonous.

Author Response

(The authors gave the same response as above.)
